# Hollow-Core Fiber-Based Biosensor: A Platform for Lab-in-Fiber Optical Biosensors for DNA Detection

**DOI:** 10.3390/s22145144

**Published:** 2022-07-08

**Authors:** Foroogh Khozeymeh, Federico Melli, Sabrina Capodaglio, Roberto Corradini, Fetah Benabid, Luca Vincetti, Annamaria Cucinotta

**Affiliations:** 1Department of Engineering and Architecture, University of Parma, Parco Area delle Scienze 181/A, 43124 Parma, Italy; foroogh.khozeymeh@unipr.it; 2Department of Engineering “Enzo Ferrari”, University of Modena and Reggio Emilia, Via Vivarelli 10, 41124 Modena, Italy; federico.melli@unimore.it (F.M.); luca.vincetti@unimore.it (L.V.); 3Department of Chemistry, Life Sciences and Environmental Sustainability (SCVSA), University of Parma, Parco Area delle Scienze 17/a, 43124 Parma, Italy; sabrina.capodaglio@unipr.it (S.C.); roberto.corradini@unipr.it (R.C.); 4CNRS UMR 7252, XLIM Research Institute, University of Limoges, 87000 Limoges, France; f.benabid@xlim.fr

**Keywords:** hollow-core, optical fiber, biosensor, DNA detection

## Abstract

In this paper, a novel platform for lab-in-fiber-based biosensors is studied. Hollow-core tube lattice fibers (HC-TLFs) are proposed as a label-free biosensor for the detection of DNA molecules. The particular light-guiding mechanism makes them a highly sensitive tool. Their transmission spectrum is featured by alternations of high and low transmittance at wavelength regions whose values depend on the thickness of the microstructured web composing the cladding around the hollow core. In order to achieve DNA detection by using these fibers, an internal chemical functionalization process of the fiber has been performed in five steps in order to link specific peptide nucleic acid (PNA) probes, then the functionalized fiber was used for a three-step assay. When a solution containing a particular DNA sequence is made to flow through the HC of the TLF in an ‘optofluidic’ format, a bio-layer is formed on the cladding surfaces causing a red-shift of the fiber transmission spectrum. By comparing the fiber transmission spectra before and after the flowing it is possible to identify the eventual formation of the layer and, therefore, the presence or not of a particular DNA sequence in the solution.

## 1. Introduction

Optical biosensors are gaining attention because of the increasing demand for biological and chemical analyte detection in a wide range of applications including clinical analysis [1], food quality control [2], defense and security [3], and environmental monitoring [4]. A biosensor device is made of three main elements: the biorecognition element, transducer, and receiver. There are different kinds of biosensors named electrochemical, thermometric, piezoelectric, and magnetic sensors [5,6]. It is worth mentioning that electrochemical sensors were the first commercialized sensors for glucose measurements useful for diabetic patients [7,8]. Furthermore, recently, another kind of electrochemical sensor based on field-effect transistors (FET) has shown promising results for point-of-care-based diagnosis [9]. Parallel to electrochemical sensors, optical biosensors also look promising for point-of-care-based diagnosis [1] and food quality control [2]. Optical biosensors are modern analytical devices that employ light-guiding technologies as their transducer part. Optical biosensors exploit the properties of light for bio/analyte detection. Changes in optical properties of the light in contact with the bio/chemical analytes lead to changes in the output spectrum or in the electrical signal recorded by the photodetector. Some of the advantages of optical biosensors compared with other sensors are the immunity of EM fields to electrical noises, higher sensitivity, simpler mechanism and detection protocol, more reliability, more flexibility, lower cost, and more compactness [5]. Regarding the pros of being simple, compact, and usability with untrained personnel, particularly label-free optical biosensors are attracting more attention [5]. In this scheme of detection, the bio/chemical analytes are directly detected without the laborious process of analyte molecule labeling [10,11,12]. Regarding their transducer part, optical biosensors are categorized into different groups of resonator-based [13,14,15], waveguide-based [16,17], and interferometric-devices based [18,19]. Although these optical sensors have demonstrated high sensitivities for both bulk and surface sensing measurements [20,21,22], they still suffer from a critical issue. Indeed, the integration need of photonic and fluidic technologies [23,24] envisages unified sensor devices. Photonic crystal fiber (PCF)-based biosensors are a group of optical biosensors that meet the need for effective integration between fluidics and photonics [25,26]. The integration of fluidics and photonics in these miniaturized structures may pave the way for lab-in-fiber technologies employing in vivo biosensing [24]. PCFs can be referred to as the structures which confine the light inside a core surrounded by a microstructured cladding composed of air holes running along the length of the fiber [25]. Thus, gas or liquid solutions can be infiltrated inside these air holes leading to the creation of biological layers on the dielectric inner walls of the fiber [27]. This advantage can be furthermore exploited if HC-PCFs are considered thanks to the possibility of having a strong interaction between the sample and the light guided by the fiber. HC-PCFs can rely on two different transmission mechanisms: Photonic Band-gap (PBG) or inhibited coupling (IC). The former presents disadvantages in terms of bandwidth and, moreover, this mechanism requires complicated microstructured cladding. While the latter allows covering a wider range of wavelengths. In fact, the transmission spectra of this kind of fiber are characterized by an alternating sequence of high and low transmission bands. In particular, there is high transmission when the coupling between the fundamental core mode and the cladding modes is prevented [28]. The position of the transmission bands is defined by the thickness of the microstructured cladding which can be composed of much simpler structures and larger pitches with respect to PBG fibers, allowing easier infiltration of solutions. So, HC-IC fibers represent a promising platform for the development of label-free sensors for biological molecules detection [29,30]. The formation of a biological layer due to the molecular interaction of the target analyte with the suitably functionalized glass core surface is exploited for detection because it changes the thickness of cladding that is directly translated into a shift in the transmission spectrum of the fiber without the use of additional transducers.

Aim of this study is to demonstrate, as a proof of concept, the suitability of this platform for DNA detection using this wavelength shift. For this purpose, a five-step functionalization process of the inner surface of the fiber has been performed in order to link specific peptide nucleic acid (PNA) probes able to selectively bind target DNA. Then, a three-step detection using DNA, a second PNA signaling probe and streptavidin binding can be performed. Precise monitoring of the wavelength bands in output spectra of the fiber has demonstrated a wavelength shift in the order of a few nanometers. This wavelength shift measured between the transmission fiber spectra before and after infiltration with a solution containing the streptavidin molecules, has been obtained to a significant extent in the presence of the DNA molecules.

## 2. Hollow-Core Fiber-Based Biosensor

The biosensor here proposed is based on a piece of HC-TLF with a length *L* shown in Figure 1a. The fiber cross-section is schematically shown in Figure 1b. The fiber has a hollow core with radius $R_{co}$, surrounded by eight silica tubes with thickness *t*, radius $r_{t}$ and refractive index $n_{d}$. The waveguiding mechanism of the HC-TLFs is based on the interaction between the core modes (CMs) confined in the HC and the cladding modes (CLMs) confined inside the silica tubes [28]. According to the coupled-mode theory, power exchanging or coupling between the modes depends on the phase-matching and the amount of overlap integral between the modes [29]. In ICFs, the latter strongly depends on the transverse spatial oscillations of the CLMs. CLMs with slow spatial oscillations give high overlap integral with CMs and make them highly lossy. CLMs with quick spatial oscillations are weakly coupled with CMs which exhibit low loss. The phase-matching condition with slow oscillating CLMs occurs at the spectral regions corresponding to wavelength close to:(1)$$\lambda_{m}=\frac{2t}{m}\sqrt{n_{d}^{2}-1}$$
being *m* and integer number. Conversely weak coupling with quick varying CLMs correspond to:(2)$$\lambda_{m+1/2}=\frac{2t}{m+1/2}\sqrt{n_{d}^{2}-1}.$$

Therefore, the spectrum of transmissivity is:(3)$$T=\frac{P_{out}}{P_{in}}$$
with $P_{out}$ and $P_{in}$ being the output and input power of the fiber, which is characterized by a succession of high and low transmission regions located, respectively, at $\lambda_{m+1/2}$ and $\lambda_{m}$. Figure 2 shows an example of *T* spectrum.

The working mechanism of sensing here discussed is based on the dependence of those wavelengths on the tube thickness *t*. This is different from PC-fiber-based biosensors based on the interaction between the core modes and analytes [27]. In the HC-TLF-based biosensor examined here, the creation of the bio-layers with thickness of the $t_{bi}$ on the tube surfaces (Figure 1c), will increase the total tube thickness causing a red-shift of the fiber transmissivity spectrum (see Figure 2). The spectral sensitivity (*S*) of the high transmissivity band edges can be expressed as:(4)$$S_{\lambda ,t_{bi}}=\frac{d\lambda }{t_{bi}},$$
where dλ is the spectral wavelength shift and $t_{bi}$ = dt/2 is half of the tube thickness variation. It is worth mentioning that, unlike the other PC-fiber-based biosensors, the spectral red-shift and, consequently, the sensitivity of this biosensor is not due to the guided mode effective refractive index changes. The formed bio-layer in the proposed biosensor here does not affect the effective refractive indices of the core modes and just affects the cladding mode parameters. More theoretical details about the estimation of this spectral shift can be found in [29]. The process of bio-layers formation on the internal surfaces of the fiber includes five steps, described in detail in the next section, and contains the bindings among the PNA, DNA, streptavidin molecules, and other chemical components used in the chemical functionalization of the HC-TLF. A schematic of the resulting bio-layer is shown in Figure 1d.

## 3. Chemical Functionalization Protocol

In order to have a selective detection of DNA and streptavidin analytes using the proposed HC-TLF, a five-step functionalization process for binding peptide nucleic acid (PNA) probes to the inner surface was used. Once the fiber was prepared, DNA detection was performed by infiltration of the sample, followed by that of a second biotinylated PNA probe and then of a streptavidin solution. These processes are shown in Figure 3. In the first derivatization step, the HC-TLF is cleaned and activated with a strong acid solution (MeOH: HCl, 1:1), in order to remove eventual residues of organic compounds and to activate the silica surface; then, the fiber is infiltrated with a solution of (3-aminopropyl) triethoxysilane (APTES, 0.1%) in absolute ethanol. This organofunctional alkoxysilane molecule allows us to have the silanization of the inner fiber surface and to have a positively charged layer, in specific the terminal amino groups that are protonated at neutral pH (Figure 3a). In the second step as can be seen in Figure 3b, we flow through the fiber a solution of succinic anhydride (0.25 M in N, N-dimethylformamide (DMF)) in order to obtain terminal acidic moiety attached to the fiber. In the third step, we follow the functionalization process with infiltration of the fiber with a solution of N, N’-diisopropyl carbodiimide (DIC), and N-hydroxysuccinimide (NHS) in DMF as a solvent (c = 0.25 M for both solutions). This is a crucial step because the reaction between these two reagents allows us to have an activated ester on the fiber surface available to react with the free amino group of the PNA solution, producing the corresponding amide bond. After this step, a solution of the PNA capture probe in DMF (30 µM concentration) in the presence of excess DIPEA (50 μM concentration) was flowed through the fiber to obtain the desired binding (Figure 3c).

In the detection steps, the functionalization procedure was changed compared with [30] in order to optimize the DNA detection process and avoid unspecific binding, by infiltration with ethanolamine in TRIS base (0.3% ETA pH:9) solution (Figure 3d), in order to block the unreacted active sites. Using these derivatized fibers, the DNA sensing protocol was performed. The fiber was flowed with complementary DNA (5 μM concentration) in PBS as a buffer for the capture of the target DNA by the formation of the stable duplex DNA: PNA (Figure 3d). Then, (Figure 3e) a solution of a second PNA functionalized with biotin (signaling probe) in DMF solution (30 μM concentration) was injected inside the fiber in order to label the DNA with biotin through non-covalent DNA: PNA interactions at the opposite end of the DNA. Finally, in the last step, infiltration of a solution including the streptavidin analytes (1 mg/mL concentration) has been performed (Figure 3f). In order to increase the thickness of the bio-layer for optical detection. The chemical properties of the PNA, DNA, and biotinylated PNA used in our experiments, are mentioned in Table 1.

The internal fluid-infiltration of the HC-TLF was performed by applying a nitrogen pressure of 2 atm to a polytetrafluoroethylene (PTFE) tubing reservoir (100 μL), connected to the terminal part of the fiber through a polyetheretherketone (PEEK) ferrule and a PTFE adapter. The flow of all the solutions through the fiber was confirmed by observing the drops of a liquid flow at the opposite end of the fiber. Furthermore, during the infiltration, the system was controlled regularly to ensure a right and precise infiltration. After each step in both the derivatization and sensing processes, the fiber was flowed with double distilled water then emptied by flowing nitrogen. Finally, we performed the optical measurements on these empty fibers. During all the steps of derivatization and sensing processes, in order to avoid environmental errors and contaminations, fiber was kept fixed in the same position as much as possible. Furthermore, all the measurements were carried out in a cleaned room.

## 4. Optical Setup

The optical setup for employing an HC-TLF as a biosensor is depicted in Figure 4. As a light source, a supercontinuum white light source with diffraction-limited light in the wavelength range of 450–2400 nm was used. The alignment of the setup includes two main steps. In the first step, the output light of the laser is coupled to the 40 cm long HC-TLF through the appropriate lens with an appropriate focal length and a 3-axis NanoMax Flexure stage. This stage with micro-metric, precise, and continuous motions along the three-axis of *X*, *Y*, and *Z* enables us to couple input light into the HCTLF-input (left side of the Figure 4). In the second step of setup alignment, the output light passed from an optical objective lens, is hit on a CCM1–PBS252/M –30 mm Cage Cube-Mounted beam splitter cube, applicable in the wavelength range of 620–1000 nm (middle of the Figure 4). One beam is guided to an ultra-compact, lightweight Zelux Camera with a visible range of wavelengths connected to a laptop for real-time monitoring of the output beam. The second beam after passing the final objective is sent to an optical spectrum analyzer (OSA) (AQ-6315A/-6315B) with a resolution bandwidth of 0.5 nm to measure the output transmission spectrum in the wavelength range of 350–1750 nm. Both the schematic and real stage of the optical setup have been shown in Figure 4. As can be seen in Figure 4 (top), another advantage of this sensor is that it does not require a coherent source of light.

## 5. Results and Discussion

Figure 5 shows the HC-TLF cross-section used in the experiment. The fiber has a core radius $R_{co}$ = 18 μm and it is surrounded by eight silica tubes with a radius of $r_{t}$ = 7 μm and thickness of *t* = 700 nm. The transmission spectrum shown in Figure 5 is acquired by the OSA before infiltrating the biochemical solutions inside the fiber. The transmission spectrum includes three high loss regions in the wavelength ranges of approximately 500–540 nm, 729–805 nm, and 1277–1645 nm. Precise monitoring of the spectrum shift is obtained by monitoring the wavelengths corresponding to the sharp transition from high and low *T*.

Figure 6 compares the transmission spectra before (biotinylated PNA) and after (streptavidin) bio-layer binding (steps e and f in Figure 3, respectively). In order to test the reliability of the results, the process has been repeated three times, as shown by (1), (2), and (3) in Figure 6. A red-shift of the transitions in the first, second and third regions is observed. As it can be seen in Figure 6, the wavelength shifts range from 1.83 nm to 3.81 nm for the front, from 2.84 nm to 5.76 nm, and from 9.82 nm to 17.92 nm, are measured at a reference signal level of −35 dB, respectively, in the first, second, and third loss regions.

Furthermore, at other reference signal levels (from −35 dB to −50 dB), the average wavelength shift between the graphs can be observed and measured. In Figure 7, the wavelength shifts between the transmission spectra of the fiber before and after infiltration of the solution containing the streptavidin molecules, are measured. These values have been measured at, respectively, three different wavelengths of 505 nm, 735 nm, and 1290 nm. The obtained results are comparable with the reported values in other works [22,31], where detection of Biotin-Streptavidin binding layers is performed.

As the final remark, it has been mentioned that the measured red-shifts which can be referred to as the detection of DNA and streptavidin molecules are comparable to the size of the protein molecules. It is worth mentioning that the protein molecules detected here have a refractive index of 1.45 RIU. However, the proposed biosensor can be applied for the detection of molecules such as breast cancer molecules with an RI of 1.401 [32]. In this case, another functionalization process will need to be taken. Furthermore, the red-wavelength shift observed in three wavelength bandwidths shows the flexibility of the proposed biosensor alongside usages of different light sources. Since the shift observed in the case of the second and third bands was significant (5.76 and 17.92 nm, respectively), according to the 3-sigma criterion, though we did not perform a calibration curve to determine the limit of detection (LOD), we can state that the LOD should be lower than the concentration used in the test (5 μM). We envisage that the method can be used for quantitative measurement of the DNA content, since a different concentration of DNA will lead to a different degree of ‘capture’ of the target DNA and, therefore, to the capture of streptavidin to a different extent, leading to a different shift, as previously observed with a similar format on Bragg-grated PCF using gold nanoparticles as enhancers [33].

## 6. Conclusions

An HC-TLF-based biosensing platform suitable for the detection of the DNA molecules was demonstrated. The red-wavelength spectral shifts before and after the formation of the protein molecule layers prove the detection of the DNA molecules. A five-step functionalization process of the fiber’s internal surfaces has been developed for increasing the sensitivity and the reliability of the detection. In particular, infiltration of the fiber with ethanolamine in TRIS base (pH:9) solution, has enhanced the possibility of DNA binding and final detection of streptavidin. The use of streptavidin in the final step makes the bio-layer thicker, and thus, easier the detection. Despite plenty of infiltrations, no inner fiber structure breaking has been observed. This guarantees the resilience of the internal miniaturized structure of the fiber for bio/chemical applications in fluid environments. The reliability of the results has also been confirmed throughout the multiple repetitions of the infiltration process and optical measurements. The wavelength shifts of 1.83–3.81 nm, 2.84–5.76 nm, and 9.82–17.92 nm in three wavelength ranges of, respectively, 500–540 nm, 729–805 nm, and 1277–1645 nm were measured. This reveals the flexibility of the HC-TLF-biosensor in biomolecule detection in a wide range of applications and its compatibility with numerous optical components and laser sources. Other advantages of the examined biosensor are the simple optical measurements, robustness to the excitation mode, and free from any extra reinforced techniques for molecule detection. This biosensor with an unprecedented light-waveguiding mechanism and operating principle, though only tested using standard pure samples, could show promising results for ultra-sensitive detection of the thinner bio-layers including breast cancer molecules in the lab-in-fiber platform. Further studies to assess sensitivity to target interfering biomolecules are needed to fully develop the present method into analytical or clinical applications. 

## Figures and Tables

**Figure 1 sensors-22-05144-f001:**
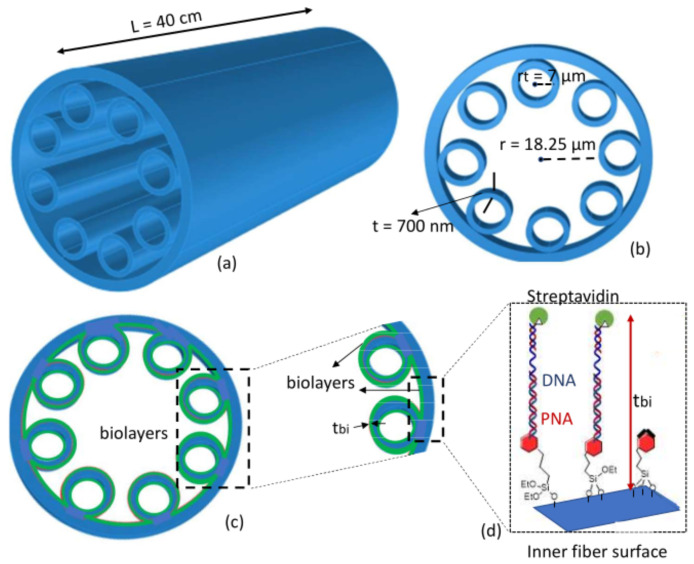
Schematics of (**a**) HCTLF-based biosensor with *L* = 40 cm, (**b**) fiber cross-section view with structural parameters, (**c**) Sensing mechanism and formation of the bio-layers inside the fiber surfaces, and (**d**) details about formation of the bio-layers, bioreceptors and streptavidin analytes.

**Figure 2 sensors-22-05144-f002:**
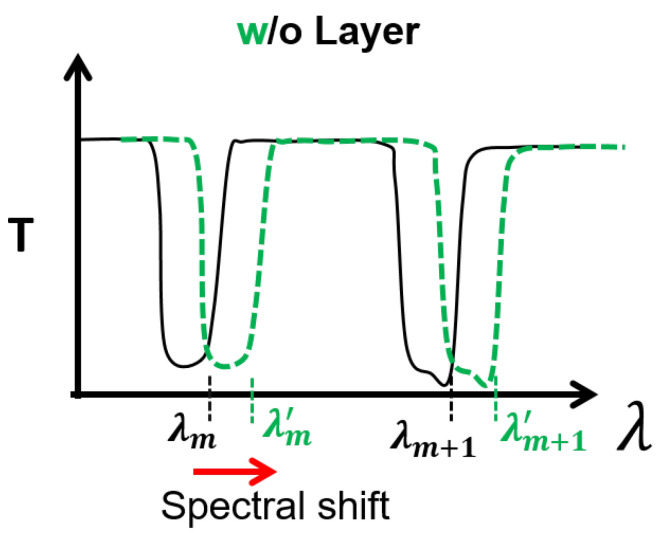
The sketched *T* spectrum shift due to the bio-layer binding of the inner surface of the fiber. The presence or absence of bio-layer has been shown, respectively, with *w* (green) and without *o* (black).

**Figure 3 sensors-22-05144-f003:**
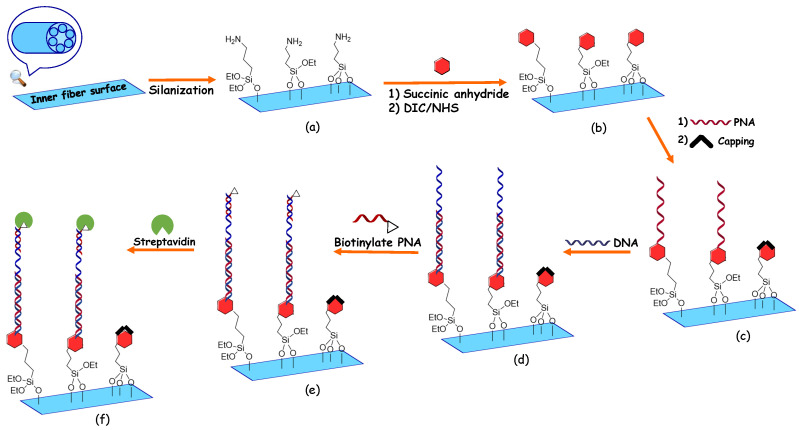
Derivatization scheme used for the HC-TLF: (**a**–**c**): derivatization; (**d**–**f**) DNA detection. (**a**) Silanization, (**b**) Reaction with succinic anhydride and activation of the terminal carboxylic moiety, (**c**) Coupling with the PNA through the terminal amino group, and blocking the unreacted active sites, (**d**) Infiltration of the HC-TLF with DNA complementary for the formation of the duplex PNA:DNA, (**e**) Flowing the biotinylated PNA signaling probe, and (**f**) infiltration of a streptavidin solution for signal magnification.

**Figure 4 sensors-22-05144-f004:**
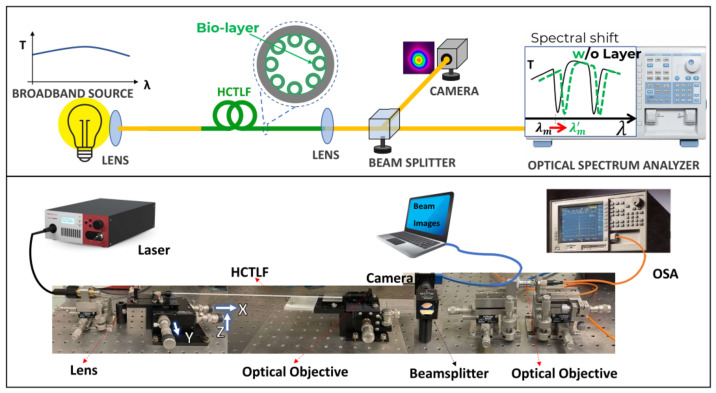
(**Top**) The schematic setup of the experiment includes a broadband coherent or incoherent source of light coupled at one end of the fiber. The transmitted light is collected in the output end of the fiber and finally, its spectrum is analyzed. The presence or absence of a spectral shift corresponds to the presence or absence of the DNA sequence searched for which have been shown, respectively, with (W) and without(O) bio-layer, (**bottom**) Image of optical setup including the laser, systems of lenses, camera, mechanical stages (with the possibility of micrometric movements along the three axes of X, Y, and Z), for guiding the laser light beam, and OSA connected to the camera is employed for recording the spectrum.

**Figure 5 sensors-22-05144-f005:**
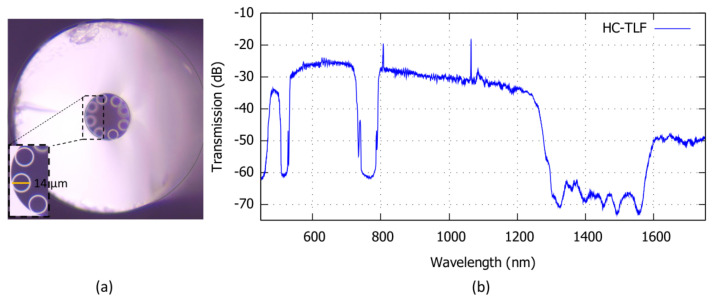
(**a**) Optical microscope image of HC-TLF cross-section (inset: magnification of the inner structure), (**b**) Transmission spectrum of the HC-TLF.

**Figure 6 sensors-22-05144-f006:**
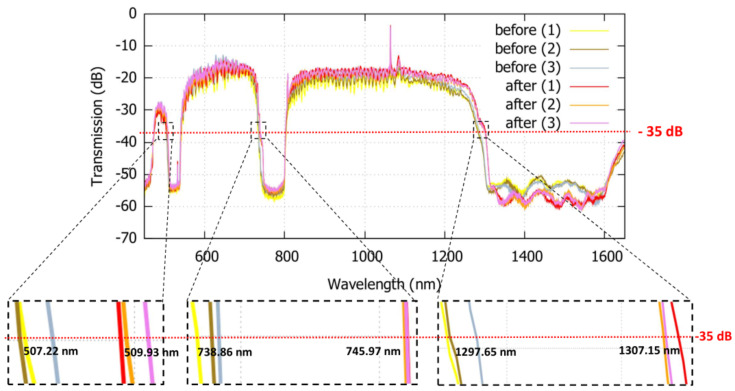
Transmission spectra of HC-TLF before and after infiltration with streptavidin solution, insets show the zoomed images of the high loss regions at the reference level of −35 dB. For both the before and after infiltration, optical measurements repeated three times, shown by 1, 2, and 3.

**Figure 7 sensors-22-05144-f007:**
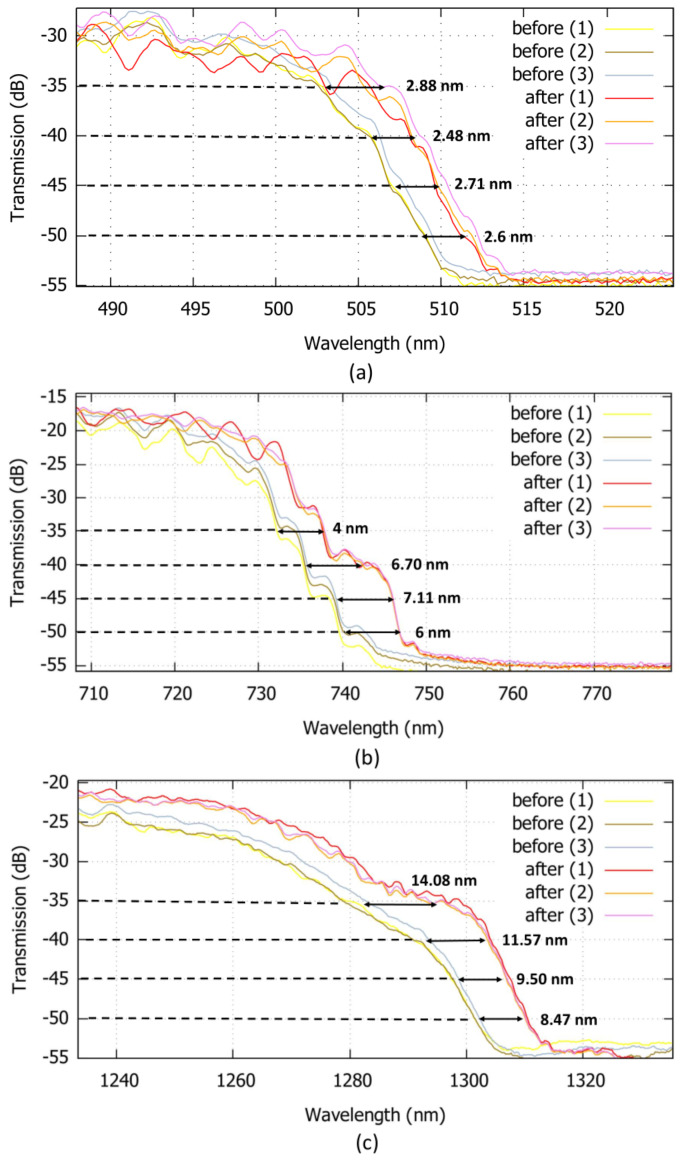
Zoomed images of the high loss regions at the reference level of −35 dB, −40 dB, −45 dB, and −50 dB, in different wavelength ranges of (**a**) 500–520 nm, (**b**) 720–750 nm, and (**c**) 1290–1320 nm. The results related to the first (1), second (2), and third (3) measurements before and after streptavidin solution infiltration, are observable.

**Table 1 sensors-22-05144-t001:** Different summarized Oligo chemicals used in the chemical infiltration process.

Oligo	Name	Sequence
PNA	PNA Soy RR	N${}_{term}$ H-O-TGC TAG AGT CAG CTT-NH${}_{2}$ C${}_{term}$
DNA	50-mer SOY	5′- ACC CTA ATC ATT TCA TTT GGA GAG
	RR	GAC ACG CTG ACA AGC TGA CTC TAG CA -3′
Biotinylated PNA	PNA 8 mer-Biotin	N${}_{term}$ Biotin-O-TGG GAT TA-Gly-NH${}_{2}$ C${}_{term}$

## Data Availability

Not applicable.

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
