# Peer review of "Hollow-Core Fiber-Based Biosensor: A Platform for Lab-in-Fiber Optical Biosensors for DNA Detection"

_sensors, 2022, doi:10.3390/s22145144_

Round 1

Reviewer 1 Report

This paper use the hollow-core tube lattice fibers (HC-TLFs)  as a label-free biosensor for the detection of DNA molecules. The HC-TLFs may be difficult to be prepared for production.

1. PNA‘s full name?

2.The authors should consider the methods about the welding between the HC-TLFs and the normal optical fiber when in practical use in future.

3. Bio-analyte for DNA detection always impurity contains, the authors may explore the impurity effects to the biosensor system.

Author Response

We would like to thank the reviewer for the constructive comments and
recommendations. We have modified the paper accordingly, the revised parts
are bold-texted in the new manuscript (MS).
Our responses regarding the reviewer’s comments and recommendations
have been expressed in the attached file (Revision report+revised paper with bold-texted changes+revised paper without the bold-texted changes).

Reviewer 2 Report

In this paper, a hollow-core tube lattice fiber is proposed as a label-free biosensor for the detection of DNA molecules. A solution containing a particular NDA sequence is made to flow through the HC of the fiber, a bio-layer is formed on the cladding surfaces causing a red-shift of the fiber transmission spectrum. Some experiments are carried out. However, there are still many questions for the manuscript’s revision.

(1) Can this method be as a quantitative measurement tool for DNA? If not, how to improve the method to meet this requirement?

(2) The author claimed that the red-shift of the wavelength is due to the change of the thickness. I am wondering whether it is an effective refractive index. 

(4) I suggest the author can give theoretical analysis about the spectrum output at different effective refractive index.

(5) As a sensor, it is important to carry out the calibration for the sensor. Is the calibration done?

(6) How about the sensitivity of the sensor for detecting DNA? More information should be provided.

(7) How about the effect of the length? More information should be provided.

(8) Shown in Fig. 5, there are some reducing areas. It is better to give some explanation.

(9) Red-shift is observed at the falling edges. I am interested that how about the rising edge? There is the same phenomenon? If not, what’s the reason? 

Author Response

(The authors gave the same response as above.)

Reviewer 3 Report

Comments:

The authors presented an HC-TLF-based biosensor for the detection of the DNA molecules. The Idea looks interesting. I will be happy to recommend it.  Before publishing this manuscript, the authors need to solve the following issues….

1.      Performance comparison with other reported works.

2.      How did they validate their experiment? I could not find any clinical sample/spiking sample test.

3.      Selectivity interference study must need to include. Also, control experiment

4.      Fig. 3 I could not see any blocking step after antibody/PNA immobilization? Then how did they overcome the non-specific binding of DNA Solution?

5.      How did the authors select the perfect parameters? Did they perform any optimization for any steps?

6.      What is the limit of detection and sensitivity?

7.      Why did they perform 8-step functionalization? more discussion is needed, also in the introduction part is poorly written.  What exact existing problems they overcome in this research? 

Author Response

(The authors gave the same response as above.)

Reviewer 4 Report

 In this manuscript, the authors studied a platform for lab-in-fiber-based biosensors and to achieve DNA detection by using these fibers, an internal chemical functionalization process of the fiber has been performed in eight steps.  When a solution containing a particular DNA sequence is made to flow through the HC of the TLF, a bio-layer is formed on the cladding surfaces causing a red-shift of the fiber transmission spectrum.

 Although, the results look promising, but there are some comments that should be addressed before publishing in Materials.

1-    The abstract of the manuscript needs to be written in the more precise and descriptive way which clearly describe the major finding of this research.

2-    The labels of the figures are written is the same way. All the Figures should be marked/labels in the same way. (e.g., Figure 5 and Figure 7 has their labels a, b, c at different places).

3-    The most recent report on the biosensors should be highlighted in the introduction on an appropriate place as (Nanomaterials 202212(8), 1305; https://doi.org/10.3390/nano12081305),

4-    The English language need to improve, and some minor typo errors could be improved too.

Remarks: Published after solving all the above comments/concerns.

Author Response

(The authors gave the same response as above.)

Round 2

Reviewer 2 Report

The author has considered most of the questions.

Reviewer 3 Report

Authors addressed my concerns accordingly.  It can be accepted in the current form. 

Reviewer 4 Report

Authors have modified the manuscript as per my sugeestion so i agree to accept this manuscript.